# Estimating the number and percentage of children who experience parental incarceration in Canada using whole population administrative and vital statistics data

Fiona G. Kouyoumdjian[1]*, Martha Paynter[2]*, Else Marie Knudsen[3], Ruth Croxford[1], Susan J. Bondy[4], Lindsay Jennings[1], Nancy Russell[5], Raya Semeniuk[6], Christine Bentley-Wang[7], Amanda Butler[8], Tenzin Butsang[1,4], Nicole L.A. Catherine[9], Alice Cavanagh[10], Jennifer Leason[11], Jessica Liauw[12], Kate McLeod[1], Natalie Owl[1,13], Akwasi Owusu-Bempah[14,15], Salony Sharma[16]

1 Department of Family Medicine, MacMaster University, Hamilton, Ontario, Canada, 2 Faculty of Nursing, University of New Brunswick, Fredericton, New Brunswick, Canada, 3 Department of Social Work, Trent University, Peterborough, Ontario, Canada, 4 Dalla Lana School of Public Health, University of Toronto, Toronto, Ontario, Canada, 5 Canadian Coalition for Children with Incarcerated Parents, Etobicoke, Ontario, Canada, 6 Native Women's Association of Canada, Gatineau, Quebec, Canada, 7 Department of Pediatrics, McMaster University, Hamilton, Ontario, Canada, 8 Department of Criminology, Simon Fraser University, Burnaby, British Columbia, Canada, 9 Faculty of Health Sciences, Simon Fraser University, Burnaby, British Columbia, Canada, 10 Department of Family and Community Medicine, University of Toronto, Toronto, Ontario, Canada, 11 Department of Political Science, University of Calgary, Calgary, Alberta, Canada, 12 Department of Obstetrics and Gynaecology, University of British Columbia, Vancouver, British Columbia, Canada, 13 University of Regina, Regina, Saskatchewan, Canada, 14 Department of Sociology, University of Toronto, Toronto, Ontario, Canada, 15 Centre for Addiction and Mental Health, Toronto, Ontario, Canada, 16 Michael G. DeGroote School of Medicine, McMaster University, Hamilton, Ontario, Canada

☯ These authors contributed equally to this work.
* kouyouf@mcmaster.ca (FGK); martha.paynter@unb.ca (MP)

## Abstract

### Background

There is a lack of systematic data on children who experience parental incarceration in Canada.

### Objective

To use linked data to estimate the number of children who experienced parental incarceration in five Canadian provinces from 2015 to 2021, and to describe parent and child characteristics.

### Methods

We accessed data from the Canadian Correctional Services Survey, a Statistics Canada survey that included person-level administrative data for all people incarcerated in British Columbia, Alberta, Saskatchewan, Ontario, and Nova Scotia provincial

**Data availability statement:** Access to the study dataset is limited to specific individuals with relevant approvals from Statistics Canada, and this restricted access was part of the protocol approved by the Hamilton Integrated Research Ethics Board. Additional information regarding access to the Children with Incarcerated Parents dataset (https://crdcn.ca/data/children-with-incarcerated-parents/) is available through the Canadian Research Data Centre Network at https://crdcn.ca/.

**Funding:** This study was funded by the Canadian Institutes of Health Research (CIHR): award number 488889. FK was the funding recipient. The funder had no role in study design, in the collection, analysis, and interpretation of data, in the writing of this manuscript, or in the decision to submit the paper for publication.

**Competing interests:** The authors have declared that no competing interests exist.

correctional facilities between 2015 and 2021. We identified their children using three methods: children with a person in the Canadian Correctional Services Survey listed as a parent on their birth certificate, children of people in the Canadian Correctional Services Survey who had received child tax benefits, and children of females in the Canadian Correctional Services Survey with a live birth in hospital discharge records. We calculated the number and percentage of children <18 who experienced parental incarceration, and described parent and child characteristics.

## Results

For 2015–2021, we identified 93,090 incarcerated parents of children <18 and 169,740 children <18 who experienced parental incarceration. We found that per day in the included provinces, 0.23% of children <18 in 2016 and 0.27% in 2017 experienced parental incarceration, and per year, 1.2% of children <18 in 2016 and 1.3% in 2017 experienced parental incarceration. For children who experienced parental incarceration, 87.4% had one and 12.0% had two parents who experienced incarceration during the study period, and 5.9% had at least one Black parent and 30.5% at least one Indigenous parent. Children who experienced parental incarceration had a median of 60 (IQR 11–208) and a mean of 166.5 (SD 256.5) total days of parental incarceration during the study period. For parents who experienced incarceration, 22.6% were female, 77.4% were male, 6.0% were Black, and 31.6% were Indigenous.

## Significance

This is the first Canadian study to systematically estimate the number and percentage of children who experience parental incarceration. Given data limitations, our findings of the number and percentage of children should be treated as minimum estimates. While further research is needed to fully quantify the prevalence and burden of this adverse childhood experience, these minimum estimates can be used to raise awareness of the issue of parental incarceration in Canada. Evolving evidence, including this study, is instrumental to advancing work to measure, prevent, and mitigate the harms associated with parental incarceration for children and families.

## Introduction

Parental incarceration is recognized as one of the Adverse Childhood Experiences (ACEs), which are associated with health risks across a child's life course [1,2] and have cumulative, dose-response effects [3,4]. A growing international body of evidence demonstrates that children who experience parental incarceration have worse physical health, mortality, cognitive skills and academic performance, socioemotional skills, risk behaviours, mental illness and externalizing and internalizing symptoms, material hardship, and delinquent behaviour [5–10], with some evidence of effect modification by child age, child sex, and parental sex [5,9,10]. These adverse child

health outcomes are likely both associated with and caused by parental incarceration [6]. Parental incarceration is correlated with other risk factors such as exposure to parental relationship dissolution, family violence, parental substance use, lack of healthcare access, residential instability and homelessness, racism, and poverty [9,11–14]. Further, parental incarceration may directly affect child health status through mechanisms such as impacts on material hardship and parent-child attachment, child trauma associated with parental arrest, experiences of stigma and shame including stereotyping by professionals such as teachers and counsellors [14,15], and a lack of child supervision [5,9].

Given the poor population health of children who experience parental incarceration, it is important for communities and countries to have access to valid data about children who experience parental incarceration to support interventions at structural and individual levels. However, there are currently no credible estimates of the number of children who experience parental incarceration in Canada [16,17]. Limited efforts to construct estimates from correctional admissions data have been plagued by concerns regarding validity [17], as they have relied on back-of-the-envelope calculations that inappropriately extrapolated data from a single correctional facility to the whole population of people in custody in Canada [18]. Further, while Canadian correctional facilities collect a variety of demographic information about detained and incarcerated persons at each admission, quantitative data about parenting are not collected in any consistent or accessible manner [19].

The lack of data about children who experience parental incarceration renders them statistically invisible, hindering policy and program development, and public, political, and media interest [20,21]. In addition, this data gap contravenes Canada's obligations under international human rights instruments. Canada is a signatory to the 2010 United Nations Rules for the Treatment of Women Prisoners and Non-custodial Measures for Women Offenders (known as The Bangkok Rules), which defines obligations toward the children of women in prison (and acknowledges relevance to males who are fathers) such as supporting parent-child contact and gathering data about the parental status of women in prison, "taking into account the best interests of the children" [22]. Canada has ratified the UN Convention on the Rights of the Child, which specifies obligations to collect "sufficient and reliable data on children, disaggregated to enable identification of discrimination and/or disparities in the realization of rights" [23]. Canada has also passed legislation to support implementation of the United Nations Declaration for the Rights of Indigenous Peoples, which specifies obligations to prevent discrimination, to protect the rights of children, and to support economic and social conditions [24]. This progress is particularly important given the substantial overrepresentation of Indigenous people in Canadian correctional facilities [25], which leads to the separation of Indigenous parents from their children and may contribute to social and economic marginalization as well as multigenerational trauma [26,27].

Closing the evidence gap regarding children who experience parental incarceration is widely recognized by child rights advocates and researchers as essential to advancing children's rights, well-being and justice [14,28]. The recent creation of a federal data linkage environment by Statistics Canada [29] presents an opportunity to begin to close this data gap. Leveraging this data environment, we aimed to use linked data to estimate the number of children who experienced parental incarceration in five provinces in Canada from 2015 to 2021, and to describe parent and child characteristics.

## Methods

### Study design

We conducted a whole population descriptive study for the period of April 1, 2015 to December 31, 2021. We chose this period because data on parental incarceration were available only as of April 1, 2015 and data on parent and child death were available only until December 31, 2021.

### Data sources and variables

We accessed data from the Canadian Correctional Services Survey [30], which is a Statistics Canada survey that included, at the time of dataset creation, person-level administrative data for persons incarcerated from April 1, 2015 to

March 31, 2022 in British Columbia, Alberta, Ontario, and Nova Scotia provincial correctional facilities and from April 1, 2017 to March 31, 2022 for Saskatchewan. People on remand awaiting trial or sentencing and people sentenced to less than two years in custody are detained or incarcerated in provincial correctional facilities, while those sentenced to two or more years in custody are transferred to federal prisons. We accessed Canadian Correctional Services Survey data including admission and release dates (since April 1, 2015), self-reported racial identity (except for Saskatchewan, for which these data were not available), self-reported Indigenous identity, and sex.

We used three datasets to identify people in the Canadian Correctional Services Survey who were parents, and to identify their children. We used the Canadian Vital Statistics Births database to identify children with a person in the Canadian Correctional Services Survey listed as a parent on their birth certificate, which we consider as an indicator of being a *biological parent*. We also used the Canadian Institute for Health Information's Discharge Abstracts Database to identify parents in the Canadian Correctional Services Survey with a live birth in hospital discharge records since April 1, 1999, which is the earliest available date, which we also considered to be an indicator of being a biological parent. We used the Canadian Child Tax Benefits database to identify people in the Canadian Correctional Services Survey who had received child tax benefits, and we accessed data in the Canadian Child Tax Benefits database on their children. We consider those receiving Canadian Child Tax Benefits to be a *social parent*, as this benefit is received by people who have a child in their care, which may include biological parents and people who are not biological parents who are in the role of primary caregiver for the child.

We used the Canadian Vital Statistics Death dataset to identify deaths for parents in the Canadian Correctional Services Survey, recognizing parental death as an additional adverse childhood experience, and for children.

We also used publicly available Statistics Canada population estimates (derived from Census data) of the number of children and adults in each province and territory [31].

## Data environment and linkage

Statistics Canada's Social Data Linkage Environment facilitates health, justice, education, and income data integration through the creation of linked datasets [29]. Datasets such as tax records, vital statistics data, and immigration data are used to create a list of unique individuals: the Derived Record Depository. Each person is assigned a unique Social Data Linkage Environment identifier. Other datasets are linked to the Derived Record Depository deterministically or probabilistically based on names, sex, date of birth, postal code, and other variables, and people with data that are missing or incomplete for two key variables (*e.g.,* name and date of birth) are typically not eligible for linkage.

Statistics Canada had previously linked all datasets used in this study to the Derived Record Depository, *i.e.,* data for people in the Canadian Correctional Services Survey, those born and parents listed on birth certificates on the Canadian Vital Statistics Births database, people hospitalized (including children delivered in hospital) in the Canadian Institute for Health Information's Discharge Abstracts Database, people receiving benefits and their children from the Canadian Child Tax Benefits, and people who died in the Canadian Vital Statistics Deaths database. Data for people in the Canadian Correctional Services Survey were linked to the Derived Record Depository probabilistically using name, date of birth, city, census metropolitan area, census subdivision, postal code, province, sex, social insurance number, and health insurance number when available, with an overall linkage rate of 97% (including data for youth not accessed in this study). Data for parents in the Canadian Vital Statistics Births database had been linked to the Social Data Linkage Environment only to the end of 2017.

Statistics Canada finalized the dataset used for this project on November 25th, 2024, and our project team accessed and analysed the data in a Statistics Canada Research Data Centre beginning on December 16th, 2024. Our project team did not have access to individual identifiers in the datasets.

## Analyses

For the analyses for parents, we included people who were 18 years or older when they experienced a period of incarceration and who were identified as a biological or social parent to a child who was < 18 years old within the study period. For

children, we included those with a recorded date of birth who were <18 years old and had one or more parents incarcerated during the study period. Using these definitions, we estimated the number of children who experienced parental incarceration for the following time periods: (i) on average per day for each of the two years for which we had full-year data from the Canadian Correctional Services Survey and linked parental data from the Canadian Vital Statistics Births database (*i.e.,* 2016 and 2017), (ii) over the year for each of those two years, (iii) for April 1st 2015- December 31st 2017, as this was the period for which parental data in the Canadian Vital Statistics Births database were linked, and (iv) for the full study period of 2015–2021. We calculated descriptive statistics on the sociodemographic characteristics of parents (*i.e.,* both biological and social parents) and children during the study period. We also calculated descriptive statistics on parental incarceration and children's exposure to parental incarceration during the study period in total days and as a percentage of time during childhood "at risk" of parental incarceration, which we calculated as the time from first parental incarceration when the child was alive to the earliest of the end of the study period or when the child turned 18 or when the child died.

We next used whole population data for the included provinces to estimate the percentage of children who experienced parental incarceration and the rate of children who experienced parental incarceration. We calculated the percentage of children that experienced parental incarceration of the total population of children <18 in the included provinces per day and per year, respectively, in 2016 and 2017, using publicly available data for the number of children <18 years old in each province for 2016 and 2017 as the denominator [31]. For 2016 and 2017 for the included provinces, we also calculated the rate of children who experienced parental incarceration per year per total population (*i.e.,* of all adults and children), which is an indicator used internationally, and we used publicly available whole population data for the population in each province for 2016 and 2017 as the denominator [31].

As required by Statistics Canada, to avoid identifying any individual all frequency counts based on the data were randomly rounded up or down to the nearest multiple of 10. Therefore, total numbers may differ from table to table and the numbers within sub-categories may not sum to the reported total number. Because percentages were calculated using the rounded numbers they may not sum to 100%.

### Approvals

This research was conducted in accordance with the Tri-Council Policy Statement regarding research involving humans (TCPS2) [19], and the study was approved by the Hamilton Integrated Research Ethics Board (#16201) and Statistics Canada. Consistent with TCPS2 standards, we did not obtain individual consent for the secondary use of the present datasets, as our use of this information is unlikely to adversely affect any individual participant, and as it would have been impossible and impracticable to obtain individual consent. All datasets used in this research were accessed and merged within the Canadian Research Data Centre Network by one researcher, and linked datasets were anonymized and aggregated before removal and storage on researcher computers. No identifiable, linkable or confidential information was removed from the data centre or stored on researchers' computers.

### Community engagement and partnership

This project was conducted in collaboration with the Canadian Coalition for Children with Incarcerated Parents, the Canadian Association of Elizabeth Fry Societies, and the Native Women's Association of Canada. Consistent with Canadian standards for research involving people who are Indigenous [19], we documented the interest and partnership of the Native Women's Association of Canada in a letter that specified we would share decision-making regarding the work and collaborate regarding the interpretation, framing, and dissemination of the findings.

In February 2025 we convened three community engagement sessions to discuss these findings with experts with relevant lived experience. We collaborated with consultants specializing in community engagement and knowledge translation to develop and conduct this work [32]. We shared information about the sessions with local and national

organizations involved in relevant work (including Indigenous organizations), and through our personal and professional networks by email and using social media. Ten people participated in a session for parents who were incarcerated within the past 5–10 years while their child was < 18 and for adults who cared for a child with a parent who was incarcerated in the past 5–10 years (*e.g.,* a non-incarcerated parent, grandparent, guardian), four people participated in a session for youth/young adults (aged 18–29) with a parent who was incarcerated in the last 5–10 years, and six people participated in a session for service providers who work with children and families that experience parental incarceration. In each 2-hour session, we presented information on the research project goals, data, and initial findings, and participants provided their feedback and insights regarding the results and discussed how the research findings should be shared. We provided an honorarium of $100 for participation in the session and $50 for preparation time ($150 in total) for anyone who participated outside of the context of their employment. A summary report from the sessions is available online [33]. We used information from the sessions to inform our interpretation and framing of the findings, as well as content in the Discussion regarding next steps for research and application of the findings.

## Results

### Inclusion of parents and children

There were 249,440 people in the Canadian Correctional Services Survey dataset. We excluded 11,260 people who could not be linked in the Social Data Linkage Environment, 3,830 people with no incarceration until after 2021, and 20 people with incarceration episodes prior to December 31, 2021 that started while they were <18 years old. This left a total of 234,330 adults who experienced incarceration between April 1, 2015 and December 31, 2021.

The data linkage identified a total of 236,470 children of parents who experienced incarceration during this period, though for some of these children the parental incarceration occurred before the child's birth, after the child's death, or after the child became an adult at age 18. From this group, we excluded 310 children with a missing date of birth, 3,200 children whose parents were excluded (as above), 3,240 children born after 2021, 48,150 children who were ≥18 by the time of the first parental incarceration, and 1,380 children who died prior to the first parental incarceration. This led to a total during the study period (2015–2021) of 180,190 children who were not 18 or older and had not died when their parent experienced incarceration: *children with parents who experienced incarceration*, of whom 169,740 had been born and were still alive when their parent experienced incarceration during this period: *children who experienced parental incarceration*, and 93,060 people who experienced incarceration and were parents of a child <18 during the study period: *parents of children <18 who experienced incarceration*.

For the 169,740 children, 34.0% (n = 57,770) were identified only in the Canadian Vital Statistics Births database (*i.e.,* were identified to have experienced parental incarceration via a biological parent only), 19.5% (n = 33,040) only in the Canadian Child Tax Benefits dataset (*i.e.,* were identified to have experienced parental incarceration via a social parent only), 46.2% (n = 78,440) in both the Canadian Vital Statistics Births and Canadian Child Tax Benefits dataset (*i.e.,* were identified to have experienced parental incarceration via a biological parent and a social parent), and 0.3% (n = 490) in the Discharge Abstracts Database only (*i.e.,* were identified to have experienced parental incarceration via a biological parent only).

### Number, percentage, and rate of children <18 who experienced parental incarceration between 2015 and 2021

As shown in Table 1, on an average day in the five included provinces, 10,841 children experienced parental incarceration in 2016 and 12,649 in 2017. Over the course of a year, 54,580 children experienced parental incarceration in 2016 and 60,590 in 2017. Over the almost three-year period for which linked parental Canadian Vital Statistics Births data were available (2015–2017), 105,670 children experienced parental incarceration, and over the almost seven-year full study period (2015–2021), 169,740 children experienced parental incarceration (as above).

Of all children <18 in the included provinces, the percentage who experienced parental incarceration per day was 0.23% in 2016 and 0.27% in 2017, and per year was 1.2% in 2016 and 1.3% in 2017. The number of children who experienced parental incarceration per year per 100,000 total population (*i.e.,* of all adults and children in the included provinces) was 228.6 per 100,000 in 2016 and 250.3 per 100,000 in 2017.

### Number of parents who experienced incarceration between 2015 and 2021

For the five included provinces, 5,900 parents experienced incarceration on an average day in 2016 and 6,610 on an average day in 2017, and 29,940 parents experienced incarceration over the year in 2016 and 32,370 in 2017. From 2015 to 2017, 58,160 parents experienced incarceration, and from 2015 to 2021, 93,090 parents experienced incarceration (as above).

Across time periods assessed, between 6.5 and 7.4 children <18 experienced parental incarceration for every 10 adults who experienced incarceration (Table 1).

### Characteristics, exposures, and outcomes of parents who experienced incarceration and had children <18 between 2015 and 2021

Of the 93,090 parents, 22.6% (n = 21,010) were female and 77.4% (n = 72,050) were male, and sex was missing for 10 parents (Table 2). Between 2015 and 2021, 47.3% (n = 44,050) of parents were incarcerated at least once in Ontario, 27.7% (n = 25,770) in Alberta, 13.1% (n = 12,200) in British Columbia, 11.3% (n = 10,500) in Saskatchewan, and 4.2% (n = 3,900) in Nova Scotia. For self-reported racial/Indigenous identity, 6.0% (n = 5,610) of parents were Black and 31.6% (n = 29,410) were Indigenous.

Most parents had multiple incarcerations over the study period: the median number of episodes of incarceration per parent while they had a child <18 years old was 2 (interquartile range (IQR) 1–3) and the mean was 2.7 (SD 2.7). The median total number of days of incarceration (across all episodes) was 52 (IQR 9–192) and the mean was 158.1 (SD 252.9). The year of first incarceration was 2015 for 28.2% (n = 26,200), 2016 for 18.8% (n = 17,500), 2017 for 16.9% (n = 15,750), 2018 for 12.6% (n = 11,750), 2019 for 10.9% (n = 10,170), 2020 for 6.6% (n = 6,160), and 2021 for 6.0% (n = 5,540) of parents.

**Table 1. Number and percentage of children who experienced parental incarceration[a] and number of parents who experienced incarceration in provincial correctional facilities in BC, Alberta, Saskatchewan, Ontario, and Nova Scotia, Canada, by period[b].**

| Period | Years | Children <18 who experienced parental incarceration | | | Adults who experienced incarcerationN | Parents[e] who experienced incarceration N | Ratio of children <18 who experienced parental incarceration to adults who experienced incarceration |
|---|---|---|---|---|---|---|---|
| | | N | % of all children[c] | Rate per 100,000 population[d] | | | |
| **1 day** | 2016 | 10,841 | 0.23% | – | 16,740 | 5,900 | 6.5: 10 |
| | 2017 | 12,649 | 0.27% | – | 17,750 | 6,610 | 7.1: 10 |
| **1 year** | 2016 | 54,580 | 1.2% | 228.6 | 79,770 | 29,940 | 6.8: 10 |
| | 2017 | 60,590 | 1.3% | 250.3 | 81,910 | 32,370 | 7.4: 10 |
| **2.75 years** | 2015-2017 | 105,670 | – | – | 149,170 | 58,160 | 7.1: 10 |
| **6.75 years** | 2015-2021 | 169,740 | – | – | 234,320 | 93,090 | 7.2: 10 |

[a]This includes biological and social parents, who were identified using data on birth certificates, hospitalization data for deliveries, and child tax benefits.

[b]Does not include data for Saskatchewan for 2015–2017.

[c]Generated using Statistics Canada population estimates of all children <18 in the included provinces as denominators.

[d]Generated using Statistics Canada population estimates of all children and adults in the included provinces as denominators.

[e]Parents were adults who experienced incarceration and had children <18 during the period under study, and includes both social and biological parents.

**Table 2. Characteristics, exposures, and outcomes of parents[a] who had children <18 and experienced incarceration in provincial correctional facilities in BC, Alberta, Saskatchewan, Ontario, and Nova Scotia, Canada between 2015 and 2021, N = 93,090.**

| Characteristic | | % of N = 93,090 (n) (unless otherwise indicated) |
|---|---|---|
| **Sex** | Male | 77.4% (72,050) |
| | Female | 22.6% (21,010) |
| | Missing | 0.01% (10) |
| **Self-reported racial/Indigenous identity** | Black | 6.0% (5,610) |
| | Indigenous | 31.6% (29,410) |
| **Year of earliest incarceration in a provincial correctional facility 2015–2021** | 2015 | 28.2% (26,200) |
| | 2016 | 18.8% (17,500) |
| | 2017 | 16.9% (15,750) |
| | 2018 | 12.6% (11,750) |
| | 2019 | 10.9% (10,170) |
| | 2020 | 6.6% (6,160) |
| | 2021 | 6.0% (5,540) |
| **Provinces in which incarcerated in a provincial correctional facility 2015-2021[b]** | Ontario | 47.3% (44,050) |
| | Alberta | 27.7% (25,770) |
| | British Columbia | 13.1% (12,200) |
| | Saskatchewan[c] | 11.3% (10,500) |
| | Nova Scotia | 4.2% (3,900) |
| **Episodes of incarceration in a provincial correctional facility 2015–2021** | Median (IQR[d]) | 2 (1-3) |
| | Mean (SD[d]) | 2.7 (2.7) |
| **Total days of incarceration 2015–2021** | Median (IQR[d]) | 52 (9-192) |
| | Mean (SD[d]) | 158.1 (252.9) |
| **In custody at the time of birth of at least one child 2015-2021** | Male (N = 72,050) | 1.0% (730) |
| | Female (N = 21,010) | 0.43% (90) |
| **Died during study period 2015-2021** | Male (N = 72,050) | 6.1% (4,420) |
| | Female (N = 21,010) | 6.9% (1,460) |

[a]Includes both biological and social parents.

[b]Parents may have been incarcerated in more than one jurisdiction during the study period.

[c]Data were available for Saskatchewan only for 2018–2021.

[d]IQR = interquartile range, SD = standard deviation.

During the study period, 90 female parents and 730 male parents were in custody at the time of the birth of at least one of their children.

During the study period, 6.3% (n = 5,880) of parents died: 6.9% (n = 1,460) of female parents and 6.1% (4,420) of male parents.

## Characteristics and exposures of children who experienced parental incarceration between 2015 and 2021

Characteristics and exposures of the 169,740 children who experienced parental incarceration while <18 are shown in Table 3. Five point nine percent (n = 10,030) had at least one parent who was Black and 30.5% (n = 62,120) had at least one parent who was Indigenous. Eighty-seven point four percent (n = 148,400) had one parent who experienced incarceration during this period, 12.0% (n = 20,410) had two parents who experienced incarceration, and 0.6% (n = 940) had three or more parents who experienced incarceration.

**Table 3. Characteristics and exposures of children <18 who experienced parental[a] incarceration in BC, Alberta, Saskatchewan, Ontario, and Nova Scotia, Canada, between 2015 and 2021,[b] N = 169,740.**

| Characteristic or exposure[c] | | % of N = 169,740 (n) (unless otherwise indicated) |
|---|---|---|
| **Parental self-reported race/Indigenous identity[d]** | At least one parent who was Black | 5.9% (10,030) |
| | At least one parent who was Indigenous | 30.5% (62,120) |
| **Number of parents who experienced incarceration 2015–2021** | 1 | 87.4% (148,400) |
| | 2 | 12.0% (20,410) |
| | ≥3 | 0.6% (940) |
| **Child age at first parental incarceration 2015–2021** | ≤4 years | 32.6% (55,090) |
| | 5-9 years | 30.4% (51,410) |
| | 10-14 years | 24.9% (42,160) |
| | 15-17 years | 12.1% (20,400) |
| **Number of episodes of parental incarceration 2015–2021** | 1 | 44.4% (75,370) |
| | 2 | 19.3% (32,690) |
| | 3 | 11.0% (18,750) |
| | ≥4 | 25.3% (42,930) |
| **Days of parental incarceration across all incarceration episodes 2015–2021** | Median (IQR[e]) | 60 (IQR 11–208) |
| | Mean (SD[e]) | 166.5 (256.5) |
| **Days during the study period (2015–2021) for which children in the dataset were alive and <18** | Median (IQR[e]) | 2,467 (2,047−2,467) |
| | Mean (SD[e]) | 2,167.0 (536.3) |
| **Percentage of childhood <18 during the study period (2015–2021) with one or more parents incarcerated** | Median (IQR[e]) | 3.0% (0.6-10.3) |
| | Mean (SD[e]) | 8.3% (12.9) |

[a]Includes both biological and social parents.

[b]Data were available for Saskatchewan only for 2018–2021.

[c]Incarceration data are only for incarceration in provincial correctional facilities and do not include incarceration in federal prisons.

[d]We do not have data for the race/Indigenous identity of the child.

[e]IQR = interquartile range, SD = standard deviation.

Following birth, the age at first parental incarceration was ≤4 years for 32.6% (n = 55,090) of children, 5–9 for 30.4% (n = 51,410), 10–14 for 24.9% (n = 42,160), and 15–17 for 12.1% (n = 20,400). Most children experienced more than one episode of parental incarceration: 44.4% (n = 75,370) of children experienced one episode, 19.3% (n = 32,690) two episodes, 11.0% (n = 18,750) three episodes, and 25.3% (n = 42,930) four or more episodes. Between 2015–2021, the total number of days of parental incarceration across all parental incarceration episodes was a median of 60 (IQR 11–208) and a mean of 166.5 (SD 256.5). The number of days during the study period for which children in the dataset were alive and younger than 18 years old (*i.e.,* "at risk" of parental incarceration) was a median of 2,467 (IQR 2,047-2,467) and mean of 2,167.0 (SD 536.3). Therefore, children who experienced parental incarceration spent a median of 3.0% (IQR 0.6–10.3) and a mean of 8.3% (SD 12.9) of their childhoods during this period with one or more parents incarcerated. Six hundred sixty children (0.4%) had at least one parent in custody on their date of birth.

## Discussion

In this population-based study, we found that a large number of children (n = 169,740) in five Canadian provinces experienced parental incarceration between 2015 and 2021, with 12,649 children (on average) experiencing parental incarceration per day in 2017. The percentage of children who experienced parental incarceration per year was 1.2% in 2016 and

1.3% in 2017, and the number of children who experienced parental incarceration was 228.6 per 100,000 total population in 2016 and 250.3 per 100,000 in 2017. Over the study period of 2015–2021, the majority of children experienced multiple episodes of parental incarceration and the median total time for which children had a parent incarcerated was about 2 months, indicating substantial exposure to parental incarceration. Across time periods studied, for every 10 adults who were incarcerated, about 7 children experienced parental incarceration. Consistent with the disproportionately high incarceration rates among Black and Indigenous populations, which results from systemic discrimination, these groups were overrepresented among children and parents in the dataset.

To contextualize these findings, the numbers of children who experienced parental incarceration in 2016 and 2017, at 228.6 and 250.3 per 100,000, respectively, were substantially higher than the rate for the European Union, in which an estimated 800,000 children experience parental incarceration annually [34] in a population of 449 million (*i.e.,* a rate of 178 children/100,000 population), and lower than the rate in the USA, in which 1,473,700 children experienced parental incarceration in 2016 [35] of a total population of 323.1 million (*i.e.,* a rate of 456/100,000 population). These rates of children who experience parental incarceration mirror relative rates of incarceration [36].

This study has several potential limitations. We note several issues related to missing data and data linkage that may impact the study's internal validity: the lack of linkage beyond 2017 for parental data on birth certificates in the Canadian Vital Statistics Births database in the Social Data Linkage Environment, lack of linkage of data to the Social Data Linkage Environment because of data that are missing or incomplete (*e.g.,* parents' names on children's birth certificates) or changes across datasets and over time in key linkage variables such as name, which may be more common in correctional datasets [37,38], that a person (parent or otherwise) other than the parent who experienced incarceration may have received tax benefits (Canadian Child Tax Benefits), that the Canadian Correctional Services Survey did not include people who were incarcerated in federal prisons for the full study period, and the Canadian Correctional Services Survey data used for analyses did not include data for Saskatchewan for 2015–2017. The lack of linked parental data in the Canadian Vital Statistics Births database is likely particularly important, since about one third of children in the dataset were identified through Canadian Vital Statistics Births database data linkage only, and we recognize that the characteristics of parents and children identified using vital statistics data may be systematically different from the characteristics of children identified only using the two other methods. In addition, some parents would not have accessed the Canadian Child Tax Benefits. In combination, these factors likely led to a substantial underestimate of the number of children who experienced parental incarceration in the five provinces included in the study during the study period. We therefore present the data in this study as initial population-based estimates that should be interpreted with caution as minimum estimates, and which should be improved upon through subsequent research. In addition, we lack important information to understand inequitable impacts of parental incarceration, including data on Indigenous and racial identity for children and for their parents who did *not* experience incarceration; without these data, we are likely underestimating the numbers of Indigenous and Black children who experienced parental incarceration, and additional analyses including data for people incarcerated in federal prisons and stratified by parental sex would also be of value given the disproportionately high incarceration rates for Indigenous women [25].

Regarding interpretability, we did not have data on whether children were in the custody of or otherwise in the care of the person who experienced incarceration during the study period. This information would be valuable to support understanding of the potential impacts of parental incarceration on children. We note that a recent Canadian survey of people incarcerated in provincial women's prisons in four provinces found that 82% had children under the age of 18 years, and of these participants, 31% were the primary caregiver for their children before incarceration [39]. In addition, recognizing that parental incarceration may represent a marker of risk as well as a cause of risk for children and families, even in the absence of data on custodial arrangements for children, our findings remain important for understanding the size and sociodemographic characteristics of a population of children that may be at increased risk of various harms (as above). Finally, as noted by experts in our community engagement, these quantitative data lack important context; qualitative

research and community-engaged knowledge translation are needed to understand the experiences associated with these data and to drive change.

Of note, based on available data in the Canadian Correctional Services Survey, this study includes correctional data for only people incarcerated in provincial correctional facilities in five provinces, with incomplete data for Saskatchewan and a lack of data for people incarcerated in federal correctional facilities for the study period. As a result, we are not able to calculate the true number or percentage of children who experienced parental incarceration in Canada during the study period. However, we can use available data to generate reasonable estimates that consider both provinces and territories that were not included in the study and children of people incarcerated in federal prisons (see S1 Appendix A for details). For children per day, if we extrapolate the ratio of children who experienced parental incarceration to adults who experienced incarceration from the study (as per Table 1) to the full population that experienced incarceration per day in provincial/territorial and federal correctional facilities [40,41], that would translate to 25,822 children who experienced parental incarceration per day in 2016 (0.55% of all children) and 27,565 children per day in 2017 (0.59%). For children per year, if we estimate the total number of adults who experience incarceration per year and then again apply the ratio of children who experienced incarceration to adults who experienced incarceration from our study, that would translate to 92,840 children who experienced parental incarceration per year in Canada in 2016 (1.98%) and 94,699 in 2017 (2.01%). These estimates should be interpreted with caution due to uncertainties regarding the generalizability of the data included in the study for all of Canada, specifically regarding the number of people who experience incarceration per year and the number of children younger than 18 for each adult incarcerated across jurisdictions. Of note, we are not providing confidence intervals for these estimates as they are not drawn from a sample process, and so variance cannot be estimated as a function of random sampling. We expect to revise these estimates as additional evidence becomes available.

Additional data and information are therefore essential to improve the validity, interpretability, and contextualization of data on the number of children who experience incarceration in Canada. More comprehensive and up-to-date linked datasets, including complete linkage of parental data in the Canadian Vital Statistics Births database with data in the Social Data Linkage Environment, and the inclusion and linkage of data for people incarcerated in federal prisons and in correctional facilities in the other provinces and territories, would enable more accurate estimates at provincial, territorial, and national levels. Access to data on Canadian Child Tax Benefits receipt dates and custodial arrangements for children during the study period would allow for analysis of parenting relationships over time. Future qualitative research and primary quantitative research (*e.g.,* from longitudinal cohort studies) would support understanding and description of the burden of parental incarceration on children in Canada, including both contemporaneous and long-term impacts on child, parent, family, and community health and social outcomes, and of resilience factors [13,14,42].

Further, research should maintain an explicit focus on the inequitable impacts of parental incarceration [13], and, recognizing the sensitive nature of the data and topics under study and potential risks of stigma, should be co-developed with relevant interest holders including people with lived experience of parental incarceration and of incarceration as parents [14]. When focused on Indigenous populations, Indigenous communities should be engaged throughout the research process, and research should integrate Indigenous frameworks and observe best practices for data governance to support appropriate research conduct and the interpretation of findings in a culturally safe and strengths-based way.

Data limitations notwithstanding, the findings from this study provide a foundation for advancing efforts to prevent parental incarceration and mitigate its impacts on children and families. As minimum initial estimates, these data should be used, as noted by experts in our community engagement, to increase awareness of the issue of parental incarceration and the substantial proportion of children facing this adverse childhood experience annually in Canada. Recognizing that parental incarceration violates children's rights, particularly the fundamental rights to family life and to protecting the best interests of the child [23,43], urgent work is indicated to support children and families. Consistent with The Bangkok Rules, correctional authorities should routinely collect data on parental status to inform access to relevant programs and supports for parents while in custody. Evidence-based strategies to prevent and mitigate the impacts of parental incarceration

include laws and policies that consider the best interests of children (for example in parent sentencing decisions), facilitating parent-child contact during parental incarceration, and strengthening programs and supports for children and families who are involved or at risk of involvement with the criminal justice system [5,14,43–45]. Such work should proceed with an explicit focus on structural inequities.

## Supporting information

**S1 File. Appendix A.** Estimation of the number of children who experienced parental incarceration per day and per year in Canada.
(DOCX)

**S2 File. HiREB approval.**
(PDF)

## Acknowledgments

Paul Robinson and Taylor Small at Statistics Canada contributed to dataset linkages, supported data access, and provided information on the datasets.

## Author contributions

**Conceptualization:** Fiona G Kouyoumdjian, Martha Paynter, Else Marie Knudsen, Ruth Croxford, Susan J Bondy, Nancy Russell, Raya Semeniuk, Christine Bentley-Wang, Amanda Butler, Tenzin Butsang, Nicole L.A. Catherine, Alice Cavanagh, Jennifer Leason, Jessica Liauw, Kate McLeod, Natalie Owl, Akwasi Owusu-Bempah.

**Data curation:** Ruth Croxford.

**Formal analysis:** Ruth Croxford, Susan J Bondy.

**Funding acquisition:** Fiona G Kouyoumdjian, Martha Paynter.

**Methodology:** Fiona G Kouyoumdjian, Martha Paynter, Else Marie Knudsen, Ruth Croxford, Susan J Bondy.

**Project administration:** Fiona G Kouyoumdjian, Martha Paynter.

**Supervision:** Fiona G Kouyoumdjian.

**Visualization:** Fiona G Kouyoumdjian, Else Marie Knudsen, Ruth Croxford.

**Writing – original draft:** Fiona G Kouyoumdjian, Else Marie Knudsen, Ruth Croxford.

**Writing – review & editing:** Fiona G Kouyoumdjian, Martha Paynter, Else Marie Knudsen, Ruth Croxford, Susan J Bondy, Lindsay Jennings, Nancy Russell, Raya Semeniuk, Christine Bentley-Wang, Amanda Butler, Tenzin Butsang, Nicole L.A. Catherine, Alice Cavanagh, Jennifer Leason, Jessica Liauw, Kate McLeod, Natalie Owl, Akwasi Owusu-Bempah, Salony Sharma.

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
