## [Decision Letter · Decision Letter 0]

30 Dec 2025

Dear Dr. Paynter,

Thank you for submitting your manuscript to PLOS ONE. After careful consideration, we feel that it has merit but does not fully meet PLOS ONE’s publication criteria as it currently stands. Therefore, we invite you to submit a revised version of the manuscript that addresses the points raised during the review process.

The reviewers expressed many reservations about both the specifics of the data source and the presentation of the linkage and analysis. I am requesting that you address wholly each of the concerns that were raised in the review, in order to meet the PLOS ONE publication criteria. I am very enthusiastic about this topic, but want to ensure that the final paper is as rigorous as it can possibly be.

We look forward to receiving your revised manuscript.

Kind regards,

Andrea K. Knittel

Academic Editor

PLOS One

Journal Requirements:

4. Please be informed that funding information should not appear in the Acknowledgments section or other areas of your manuscript. We will only publish funding information present in the Funding Statement section of the online submission form. Please remove any funding-related text from the manuscript.

Reviewers' comments:

Reviewer's Responses to Questions

**Comments to the Author**

1. Is the manuscript technically sound, and do the data support the conclusions?

Reviewer #1: Yes

Reviewer #2: Partly

2. Has the statistical analysis been performed appropriately and rigorously?

Reviewer #1: Yes

Reviewer #2: I Don't Know

3. Have the authors made all data underlying the findings in their manuscript fully available?

Reviewer #1: No

Reviewer #2: No

4. Is the manuscript presented in an intelligible fashion and written in standard English?

Reviewer #1: Yes

Reviewer #2: Yes

Reviewer #1: I appreciate what the authors are seeking to do in this study by pulling together multiple data sources and use that to estimate exposure to parental incarceration. However, understanding how reliable (or unreliable) the estimate generated from this study is requires a better understanding of and description of these data sources used. Unfortunately, in the current study, the description of the key data sources used is quite thin and is therefore hard to evaluate what these data sources are, what information they cover, and what information they do not cover. Additional information is also needed regarding the successful or unsuccessful nature of linking these data sources. For example, one sentence is provided: "Canadian Correctional Services Survey data were previously linked in the Social Data Linkage Environment with a linkage rate of 97%." However, there is no detail, for instance, on how these data were linked, and what identifiers would have been used to link these data, including to incarcerated persons and their children.

2. Prior criminal justice work has described challenges in linking across administrative datasets and potential implications for analysis based on assumptions of how linking was performed:

Tahamont, S., Jelveh, Z., McNeill, M., Yan, S., Chalfin, A., & Hansen, B. (2023). No ground truth? No problem: Improving administrative data linking using active learning and a little bit of guile. PloS one, 18(4), e0283811.

Tahamont, S., Jelveh, Z., Chalfin, A., Yan, S., & Hansen, B. (2021). Dude, where’s my treatment effect? Errors in administrative data linking and the destruction of statistical power in randomized experiments. Journal of Quantitative Criminology, 37(3), 715-749.

3. Can more detail be provided on why the 11,260 people who could not be linked to the data set were excluded:

Reviewer #2: Thank you for the opportunity to review this paper. The authors use newly available linked data from Statistics Canada to present a descriptive portrait of how many children experience parental incarceration in Canada. This is an interesting paper, and one that capitalizes on unique and new data, that documents the scope of parental incarceration in Canada. The authors explain their data and measures well. They also outline their limitations appropriately.

I have some suggestions for the authors’ consideration as they move forward with this work.

First, and perhaps most importantly, the paper currently documents the number of children who experience parental incarceration but, without more consideration of the denominator (the number of children at risk of experiencing parental incarceration), it’s difficult for me to know what to make of these numbers. I understand that there are data limitations—in this case, the numbers are only applicable for five provinces in Canada, so perhaps that is why the authors don’t want to present percentages that include the whole population of Canadian children as the denominator. But why not present the percentage of children who experience parental incarceration (with the number exposed as the numerator and the number of children in the five provinces as the denominator)? That would be accurate (or as accurate as possible given the other data limitations) and allow for more context. The paper, as it stands, only begins to delve into this important “context” on page 19. My sense is that the authors would like the data to be comprehensive and representative of the population. Given that cannot be the case, it seems that being more clear about the sampling frame—the five provinces—is the way to go.

There are a few places where more details would be helpful:

• The authors write the following: “Other datasets are linked to the DRD deterministically or probabilistically based on names, sex, date of birth, and postal code.” This is a big undertaking and there can be big differences in deterministic and probabilistic matching. More details here would be helpful.

• The authors note the following: “…data on parent and child death were available only until December 31, 2021.” It is unclear to me why death data is necessary. I initially thought that it was relevant for the denominator (number of parents at risk of being incarcerated) but I don’t think that is the case.

• There are some results that don’t appear to be in a table (unless I missed something!). It would be helpful if all findings that appear in the Results section were also presented in a table.

There are some places in the manuscript that could use some editing and/or precision. The last sentence of the first paragraph, for example, is nine lines long and difficult to read. Given there is so much important information included in this sentence, the authors might consider breaking it up when possible. Similarly, there are places in the manuscript where the authors may consider more precision. For example, the abstract notes “For children, 87.4% had one and 12.0% had two parents who experienced incarceration…” This reads like 87.4% of all children had one parent incarcerated but, instead, the denominator here is “children with any incarcerated parent.” There are places like this throughout the manuscript that could use some clarity.

The takeaways and policy/practice implications of the paper could use some more clarity. The authors note, for example, that “study findings should be used urgently to prevent parental incarceration and mitigate harms for children and families.” How so? It would be helpful if the authors could note how their findings, specifically, should be used to prevent parental incarceration?

The authors are upfront about the limitations of the data and, therefore the conclusions they can draw. For example, they write the following: “As a result, we are not able to determine the true number of children who experienced parental incarceration in Canada, nor the total amount of time children were exposed to parental incarceration.” I wonder if they might consider providing an upper-bound and a lower-bound estimate of the parental incarceration for the country (using information they have to expand beyond the five provinces)? I don’t think the authors have to make this change, but it’s an idea if they wish to make it nationally representative.

**Do you want your identity to be public for this peer review?** For information about this choice, including consent withdrawal, please see our Privacy Policy

Reviewer #1: No

Reviewer #2: No

---

## [Author Response · Author response to Decision Letter 1]

5 Feb 2026

Response to reviewers

We thank the reviewers for their helpful comments. We have indicated the reviewers’ comments in italics and our responses in normal text.

Reviewer #1: I appreciate what the authors are seeking to do in this study by pulling together multiple data sources and use that to estimate exposure to parental incarceration. However, understanding how reliable (or unreliable) the estimate generated from this study is requires a better understanding of and description of these data sources used. Unfortunately, in the current study, the description of the key data sources used is quite thin and is therefore hard to evaluate what these data sources are, what information they cover, and what information they do not cover. Additional information is also needed regarding the successful or unsuccessful nature of linking these data sources. For example, one sentence is provided: "Canadian Correctional Services Survey data were previously linked in the Social Data Linkage Environment with a linkage rate of 97%." However, there is no detail, for instance, on how these data were linked, and what identifiers would have been used to link these data, including to incarcerated persons and their children.

We have added and clarified information regarding the details of the linkage process, and we created a new section in the Methods called Data environment and linkage to make this information more easily accessible. The data linkage was conducted by the Canadian national statistics agency Statistics Canada as part of an ongoing national initiative to integrate health, justice, education, and income data sources (the Social Data Linkage Environment) rather than for this study in particular. We have included information on the type of linkage and variables used to link the data from the Canadian Correctional Services Survey in particular.

2. Prior criminal justice work has described challenges in linking across administrative datasets and potential implications for analysis based on assumptions of how linking was performed:

Tahamont, S., Jelveh, Z., McNeill, M., Yan, S., Chalfin, A., & Hansen, B. (2023). No ground truth? No problem: Improving administrative data linking using active learning and a little bit of guile. PloS one, 18(4), e0283811.

Tahamont, S., Jelveh, Z., Chalfin, A., Yan, S., & Hansen, B. (2021). Dude, where’s my treatment effect? Errors in administrative data linking and the destruction of statistical power in randomized experiments. Journal of Quantitative Criminology, 37(3), 715-749.

Thank you for pointing to these papers. We have elaborated on relevant issues in the third paragraph of the Limitations, and we have added a reference in that section to the 2021 publication by Tahamont et al.

3. Can more detail be provided on why the 11,260 people who could not be linked to the data set were excluded.

We have added further information on the procedures (including requirements) for linkage to the Social Data Linkage Environment in the Methods section (first paragraph in the Data environment and linkage section), as well as relevant content in the Limitations paragraph of the Discussion (as noted above).

Reviewer #2: Thank you for the opportunity to review this paper. The authors use newly available linked data from Statistics Canada to present a descriptive portrait of how many children experience parental incarceration in Canada. This is an interesting paper, and one that capitalizes on unique and new data, that documents the scope of parental incarceration in Canada. The authors explain their data and measures well. They also outline their limitations appropriately.

We appreciate this positive feedback.

I have some suggestions for the authors’ consideration as they move forward with this work.

First, and perhaps most importantly, the paper currently documents the number of children who experience parental incarceration but, without more consideration of the denominator (the number of children at risk of experiencing parental incarceration), it’s difficult for me to know what to make of these numbers. I understand that there are data limitations—in this case, the numbers are only applicable for five provinces in Canada, so perhaps that is why the authors don’t want to present percentages that include the whole population of Canadian children as the denominator. But why not present the percentage of children who experience parental incarceration (with the number exposed as the numerator and the number of children in the five provinces as the denominator)? That would be accurate (or as accurate as possible given the other data limitations) and allow for more context. The paper, as it stands, only begins to delve into this important “context” on page 19. My sense is that the authors would like the data to be comprehensive and representative of the population. Given that cannot be the case, it seems that being more clear about the sampling frame—the five provinces—is the way to go.

Thank you for this suggestion, and we agree that data on percentages would support interpretability. We have added information for relevant time periods on the percentage of children who experienced incarceration and the rate of children who experienced parental incarceration per 100,000 population to the Results section (Table 1 and related text in the Results section), and we have provided related content in the Methods.

There are a few places where more details would be helpful:

• The authors write the following: “Other datasets are linked to the DRD deterministically or probabilistically based on names, sex, date of birth, and postal code.” This is a big undertaking and there can be big differences in deterministic and probabilistic matching. More details here would be helpful.

As described above, we have added further details regarding how Statistics Canada linked these data.

• The authors note the following: “…data on parent and child death were available only until December 31, 2021.” It is unclear to me why death data is necessary. I initially thought that it was relevant for the denominator (number of parents at risk of being incarcerated) but I don’t think that is the case.

We used data on deaths in two ways. First, we identified the number of parent deaths, recognizing that parental death is another adverse childhood experience. Second, we examined time at risk of parental incarceration and the percent of childhood with a parent incarcerated for children, for which we right censored the follow up period when children turned 18 or if children died.

• There are some results that don’t appear to be in a table (unless I missed something!). It would be helpful if all findings that appear in the Results section were also presented in a table.

We have created two additional tables in the Results to show the findings in that format.

There are some places in the manuscript that could use some editing and/or precision. The last sentence of the first paragraph, for example, is nine lines long and difficult to read. Given there is so much important information included in this sentence, the authors might consider breaking it up when possible. Similarly, there are places in the manuscript where the authors may consider more precision. For example, the abstract notes “For children, 87.4% had one and 12.0% had two parents who experienced incarceration…” This reads like 87.4% of all children had one parent incarcerated but, instead, the denominator here is “children with any incarcerated parent.” There are places like this throughout the manuscript that could use some clarity.

Thank you for these suggestions. We have revised those specific sections of the text to improve clarity and readability. In addition, we have carefully reviewed the full text and made revisions throughout to simplify and clarify the content wherever possible.

The takeaways and policy/practice implications of the paper could use some more clarity. The authors note, for example, that “study findings should be used urgently to prevent parental incarceration and mitigate harms for children and families.” How so? It would be helpful if the authors could note how their findings, specifically, should be used to prevent parental incarceration?

We have revised this content in the Significance section of the Abstract and elaborated on the implications of the study in the final paragraph of the Discussion.

The authors are upfront about the limitations of the data and, therefore the conclusions they can draw. For example, they write the following: “As a result, we are not able to determine the true number of children who experienced parental incarceration in Canada, nor the total amount of time children were exposed to parental incarceration.” I wonder if they might consider providing an upper-bound and a lower-bound estimate of the parental incarceration for the country (using information they have to expand beyond the five provinces)? I don’t think the authors have to make this change, but it’s an idea if they wish to make it nationally representative.

We have undertaken work to develop national estimates based on extrapolation of the study data to demographic and correctional data for the whole population, and we have provided information about these estimates in the Discussion and in an Appendix. Given the substantial limitations of the data and uncertainties inherent in these estimates, we think it is more appropriate to include this information in the Discussion only, rather than as part of the study Methods and Results, and as noted in the Discussion, we have not provided confidence intervals.

---

## [Editor Report · Decision Letter 1]

20 Feb 2026

Dear Dr. Paynter,

We look forward to receiving your revised manuscript.

Kind regards,

Andrea K. Knittel

Academic Editor

PLOS One

Journal Requirements:

Additional Editor Comments:

I am so enthusiastic about this paper and am eager to see it go to "print." I appreciate your detailed responses to the reviewers comments. I have identified one instance where I believe that the text that is in that response would additionally strengthen the manuscript, and one potential typographical error. Once these can be addressed, I look forward to accepting the manuscript.

1. In the response to reviewers, you mention using the death data to calculate time at risk of parental incarceration and the percent of childhood with a parent incarcerated for children. Please include this detail in the methods section around line 205 when the descriptive analysis is mentioned.

2. Should line 270 read "<18 years old" instead of "≥18 years old"? I'm not certain, but I can't make sense of it as written.

---

## [Author Response · Author response to Decision Letter 2]

24 Feb 2026

We appreciate these positive comments. We have added this content as requested. Thank you for catching this error, and we have corrected the text to say <18 years old.

---

## [Editor Report · Decision Letter 2]

26 Feb 2026

Estimating the number and percentage of children who experience parental incarceration in Canada using whole population administrative and vital statistics data

PONE-D-25-56138R2

Dear Dr. Paynter,

We’re pleased to inform you that your manuscript has been judged scientifically suitable for publication and will be formally accepted for publication once it meets all outstanding technical requirements.

Kind regards,

Andrea K. Knittel

Academic Editor

PLOS One
---

## [Editor Report · Acceptance letter]

PONE-D-25-56138R2

PLOS One

Dear Dr. Paynter,

I'm pleased to inform you that your manuscript has been deemed suitable for publication in PLOS One. Congratulations! Your manuscript is now being handed over to our production team.

Kind regards,

on behalf of

Dr. Andrea K. Knittel

Academic Editor

PLOS One